# Investigating Slit-Collimator-Produced Carbon Ion Minibeams with High-Resolution CMOS Sensors

Lennart Volz [1,*], Claire-Anne Reidel [1], Marco Durante [1,2], Yolanda Prezado [3,4], Christoph Schuy [1], Uli Weber [1] and Christian Graeff [1,5]

1   Biophysics, GSI Helmholtz Center for Heavy Ion Research GmbH, 64291 Darmstadt, Germany; c.a.reidel@gsi.de (C.-A.R.); m.durante@gsi.de (M.D.); c.schuy@gsi.de (C.S.); u.weber@gsi.de (U.W.); c.graeff@gsi.de (C.G.)
2   Institute for Condensed Matter Physics, Technical University of Darmstadt, 64289 Darmstadt, Germany
3   Institut Curie, Université PSL, CNRS UMR3347, Inserm U1021, Signalisation Radiobiologie et Cancer, 91400 Orsay, France; yolanda.prezado@curie.fr
4   Université Paris-Saclay, CNRS UMR3347, Inserm U1021, Signalisation Radiobiologie et Cancer, 91400 Orsay, France
5   Department of Electrical Engineering and Information Technology, Technical University of Darmstadt, 64289 Darmstadt, Germany
*   Correspondence: l.volz@gsi.de

**Abstract:** Particle minibeam therapy has demonstrated the potential for better healthy tissue sparing due to spatial fractionation of the delivered dose. Especially for heavy ions, the spatial fractionation could enhance the already favorable differential biological effectiveness at the target and the entrance region. Moreover, spatial fractionation could even be a viable option for bringing ions heavier than carbon back into patient application. To understand the effect of minibeam therapy, however, requires careful conduction of pre-clinical experiments, for which precise knowledge of the minibeam characteristics is crucial. This work introduces the use of high-spatial-resolution CMOS sensors to characterize collimator-produced carbon ion minibeams in terms of lateral fluence distribution, secondary fragments, track-averaged linear energy transfer distribution, and collimator alignment. Additional simulations were performed to further analyze the parameter space of the carbon ion minibeams in terms of beam characteristics, collimator positioning, and collimator manufacturing accuracy. Finally, a new concept for reducing the neutron dose to the patient by means of an additional neutron shield added to the collimator setup is proposed and validated in simulation. The carbon ion minibeam collimator characterized in this work is used in ongoing pre-clinical experiments on heavy ion minibeam therapy at the GSI.

**Keywords:** carbon ions; minibeams; CMOS sensors; particle therapy; minibeam collimator; neutron shield

## 1. Introduction

Particle minibeam radiation therapy (pMBRT) is a promising technique to further widen the therapeutic window of particle therapy [1,2]. In pMBRT, the beam is collimated or magnetically focused to thin (<1 mm) beamlets, resulting in a spatial fractionation of the dose with alternating regions of very high (peaks) and very low (valley) doses. Depending on the center-to-center distance (ctc) between adjacent minibeam peaks, the irradiated particle and the beam characteristics (minibeam preparation, energy, lateral/angular spread), the particles' scattering inside the patient/target smears out the lateral dose distribution with increasing depth in the patient, enabling a homogenous target dose coverage. Proton minibeam therapy has been demonstrated in recent pre-clinical studies to achieve better healthy tissue sparing compared to treatments with regular 'broad' beams [3–5], improved immunoresponse [6], and to reduce adverse effects [7].

Compared to protons, heavier ions, such as helium, carbon, and oxygen [8,9], due to the reduced scattering, yield a greater PVDR for the same peak dose and at equal ctc. This advantage also increases with increasing depth in the patient. In addition, heavier ions due to their comparatively high linear energy transfer (LET) offer increased radiobiological effectiveness (RBE) compared to protons or photons. In the case of carbon ions, the point of highest RBE also coincides well with the point of highest physical dose [10]. However, the RBE is also elevated in the entrance region when compared to lighter ions. The tissue sparing achieved by ion minibeam radiation therapy (iMBRT) may further improve the ratio between the RBE in the spread-out Bragg peak compared to that in the entrance region for carbon ions. The first results from a pre-clinical experiment irradiating a target in a rabbit's brain at the Brookhaven National Laboratory with 300 µm carbon ion minibeams, presented by Dilmanian et al. [11], indicated favorable non-targeted tissue sparing capabilities. iMBRT may even be a viable route to bring back ions with $Z > 8$ for therapy, such as neon or argon, which, despite initial tests at Lawrence Berkley National Laboratory, were abandoned due to an observed increase in healthy tissue toxicity [12].

iMBRT in general can be accomplished with similar infrastructure to proton minibeam therapy, either by placing a minibeam collimator (MBC) in front of the patient [8] or by magnetically focusing the beam to achieve the desired sub-millimeter beam width [13]. While magnetically focused medical carbon ion beams would be the more elegant solution compared to collimators, such beams are not yet available. Pre-clinical experiments for iMBRT, therefore, are often limited to MBC-generated minibeams. To ensure the accuracy of such experiments, special care needs to be taken in correctly aligning the MBC with the beam [14]. For heavy ion beams, the production of secondary particles generated inside the MBC requires further attention. A previous study by Martinez-Rovira et al. [8] characterized the dosimetric quality of carbon and oxygen minibeams generated with a tungsten MBC, investigating the PVDR and the output factor (ratio between the irradiated dose with and without the collimator). In their work, carbon ion mininibeams achieved a PVDR as high as 60 at the entrance of a water phantom, which was placed 50 mm distant to the MBC exit. Their work also confirmed the favorable output factor for heavier ions in comparison to protons as a result of the reduced scattering. In a simulation study, Gozalez, Peucelle, and Prezado [15] investigated the production of secondary fragments in carbon and oxygen iMBRT with an MBC. They found that the key contribution to the valley doses stems from secondary particles, with neutrons and protons contributing most to the healthy tissue doses, while heavier fragments dominated the valley doses in the target region.

In the present work, detailed experimental and simulated investigations of secondary production and robustness analyses are provided for a brass MBC that is currently used at the GSI Helmholtz Center for Heavy Ion Research GmbH medical experiment room [16] for pre-clinical biological response measurements. Experimental characterization of the brass MBC was performed with clinical carbon ion beams at the Marburg Ion-Beam Therapy Center (MIT), employing a high-resolution CMOS monolithic active pixel sensor, the MIMOSA-28 [17] sensor originally developed for the STAR experiment at RHIC. CMOS sensors have been presented in the literature to be useful tools for X-ray minibeam experiments [18] and recently also for pMBRT [19]. The MIMOSA-28 sensor employed in our study has already been characterized for various incident ion beams [20], demonstrating its capability of discerning secondaries and primaries based on their energy loss in the sensor. Moreover, the sensor's high spatial resolution enables precise investigation of a beam's fluence profile [21], which is valuable to determine the scattering effects of the MBC slit edges. In addition to the experimental characterization, this work provides an extended investigation of the relevant parameter space for the MBC and impinging beam with additional Monte Carlo simulations. A novel shielding concept is introduced to reduce the dose from neutrons in heavy ion minibeam applications, and it is validated in simulation.

## 2. Methods

### 2.1. Experimental Setup

Measurements were conducted with carbon ion pencil beams at the MIT. For all experiments, the beam's nominal energy was set to 180.32 MeV/u. The beam focus at isocenter was well described by a Gaussian distribution with 11.8 mm FWHM in vertical direction, as measured with the CMOS sensor setup (see Section 2.1.1). In horizontal direction, the beam was best described by a double Gaussian distribution with means of 3.97 mm and $-2.23$ mm, and standard deviations of 3.7 mm and 2.8 mm, at a relative contribution of 0.45% and 0.55%, respectively. The particle rate was set to $\leq 5$ kHz to allow single-event measurements. A brass MBC of size 70 mm $\times$ 70 mm $\times$ 30 mm (width $\times$ height $\times$ thickness) was investigated. The collimator was manufactured by the GSI in-house workshop and consisted of a $\sim 1$ cm thick brass frame that held a set of 14 brass slabs forming 15 slits of 600 µm width and 50 mm height each with a ctc of 3.5 mm. The front of the collimator was positioned right at isocenter, at 101.9 cm distance from the beam nozzle exit window. The collimator center was initially aligned by eye with the in-room laser system.

To investigate the impact of MBC positioning uncertainty on the production of secondary fragments, the MBC was mounted on a setup comprising a rotational axis. The rotational axis (M-060, Physik Instrumente, Karlsruhe, Germany) has a bidirectional accuracy of $\pm 0.017°$. A set of measurements with different MBC positioning was acquired, from which the best alignment was determined.

### 2.1.1. CMOS Sensors

Single-event measurements were acquired with a setup comprising 6 MIMOSA-28 CMOS sensors. The MIMOSA-28 sensor [17] is a monolithic active pixel sensor (MAPS) with an active area of $19.9 \times 19.2$ mm$^2$ (W $\times$ H) and a pixel size of 20.7 µm. The thickness of the sensor is 50 µm, comprising a 6 µm SiO layer, a 14 µm sensitive epitaxial layer, and a 30 µm substrate layer. The sensor features line-by-line readout with a 186.5 µs integration time, i.e., a 5 kHz frame rate.

After a charged particle hit, several pixels will fire (denoted a cluster) due to the diffusion and coulomb expansion of the released ionization electrons in the epitaxial layer. The minimum energy threshold for a single pixel to fire is determined by a user threshold on the readout, which was set to six times the sensor noise (38 electrons), coming to 820.8 eV at 3.6 eV per electron production. Considering the sensor thickness, the minimum average LET for a particle crossing the sensor would need to be 55 eV/µm in silicon or 23.8 eV/µm in water to be detected. For analysis, the raw MIMOSA-28 data were first processed with the Qapivi analysis software [22], based on ROOT [23] and Geant4 [24–26] libraries. The software reconstructs the clusters from the sensor data and their hit position is determined as the cluster's center-of-mass. Events on the first and last three sensor layers were fitted by straight lines to form tracks, respectively, and the resulting front and back tracks were then connected by extrapolation.

### 2.2. Monte Carlo Simulations

In order to verify the experiments and to further investigate the performance and robustness of the MBC, Monte Carlo simulations were performed with the Geant4 particle transport toolkit version 10.7 [24–26]. Two distinct simulation setups were considered. One corresponded to the experimental setup at MIT; the other considered a simplified geometry with the MBC and a water tank for dosimetry. For all simulations, the Geant4 reference physics list "QGSP_BIC_HP_EMZ" was active. The MBC was loaded as a triangulated mesh with the CADmesh framework [27] and modelled as brass with density 8.4 g/cm$^3$. This enables flexible collimator design studies in future works. The simulation code is openly available at git.gsi.de/lennart.volz/minibeams (accessed on 8 May 2023).

### 2.2.1. Experiment Setup

The simulation world was composed of G4_Air and included the nozzle, the front scintillator, the six MIMOSA-28 sensors, and the MBC. All components were placed according to Figure 1. No specific step limit was applied in the simulations since the small sensor geometry itself enforces a very small step limit in the scoring region. Production cuts for electrons were set to 0.01 mm; for all other secondary particles cuts were set to 0.5 mm. Per setup, $10^7$ primaries were generated.

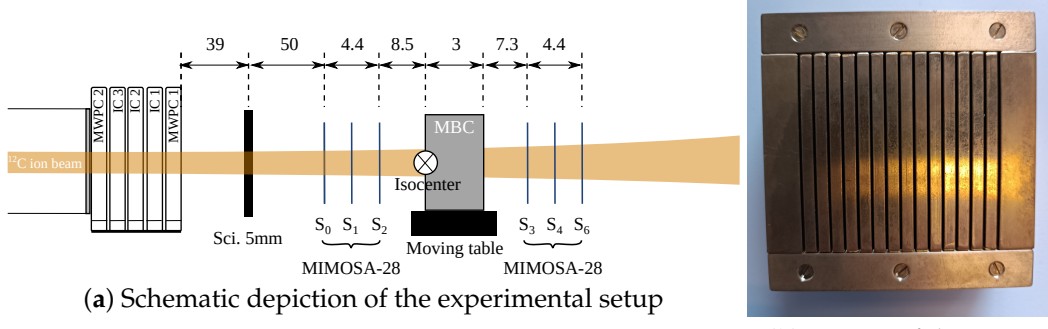

(**a**) Schematic depiction of the experimental setup

(**b**) Picture of the MBC

**Figure 1.** (**a**) Experimental setup at the Marburg Ion-Beam Therapy Center. The mini beam collimator (MBC) was positioned on a moving platform such that its front coincides with the isocenter and the slit direction was vertical. Three MIMOSA-28 sensors were positioned in front and behind the collimator, respectively. A 5 mm scintillator was positioned 39 cm after the beam nozzle exit window for a different experiment. (**b**) Picture of the employed brass MBC.

The primary carbon ions were generated right before the beam nozzle in air. In vertical direction, the beam was modeled as a single Gaussian, while, in the horizontal direction, a double Gaussian was assumed as per the measured beam profiles. For the vertical direction, the SCATTMAN [28,29] transport code was used to estimate the lateral beam spread and angular divergence before the nozzle detectors from the measured data on the CMOS sensors. The same beam divergence as in vertical was also assumed for the horizontal direction, although, here, the parameters of the double Gaussian distribution describing the lateral beam spread were taken directly from the measurements. The beam kinetic energy was set to 180.32 MeV/u with an initial 0.1% Gaussian spread. The nozzle detectors were simulated as a water slab target of 316 mm thickness and density of 0.0054 g/cm$^3$ to achieve the measured scattering properties. The end of the water slab target was placed such that its exit had a distance of 101.9 cm from the isocenter.

The MIMOSA-28 sensors were simulated as a sandwich structure of three layers, corresponding to the sensitive and insensitive volumes of the chip. In each sensitive layer, the energy deposit and the hit lateral position were scored directly from the simulation condensed history steps. The sensor's pixels were not considered in the scoring, and the cluster sizes were estimated directly from the parametrization provided by [20].

### 2.2.2. Expanded Setup

In order to provide broad tests of the parameter space, an additional setup was generated that considered a simplified version of the experiments. This setup comprised only the collimator, followed in beam direction by a water tank of 70 mm × 70 mm × 300 mm volume, in which the dose was recorded using a Geant4 primitive dose scorer. The bin size of the primitive scorer was 0.1 mm in horizontal direction, 70 mm in vertical direction, and 0.5 mm in beam direction, which also limited the step size to 0.5 mm in beam direction and 0.1 mm orthogonal to the MBC slit direction in the water tank. Production cuts were set to 0.05 mm for all particles. This cut value was chosen to obtain resilient PVDR values at manageable simulation run time. While even finer cuts may further improve the dose distribution accuracy, the difference is not expected to affect the conclusions of this work.

Similar as in previous works for protons [14], we varied the distance between MBC exit and water tank entrance to study the effect of the broadening in air on the slit dose profiles. Additionally, we investigated different beam angular divergences between 0 mrad and 4 mrad.

Furthermore, we assessed whether the MBC design would be robust against chamfered slit edges. Due to the current manufacturing process, the slit edges of the MBC can be slightly chamfered. In order to verify the use of the MBC for future experiments, the effect of this on the dose profile following different edge scattering was studied. For this, a 1 mm chamfer was applied to all the edges in the collimator stl. The chamfered collimator is shown in Figure 2.

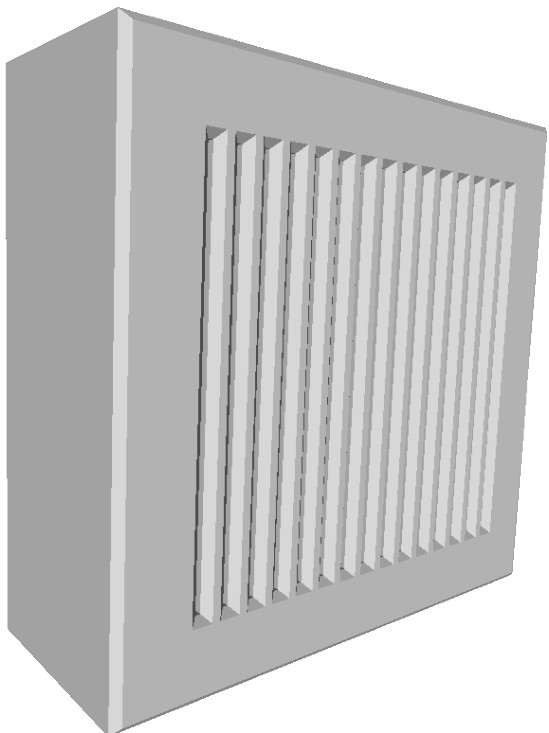

**Figure 2.** Schematic drawing of the collimator with chamfered slit edges.

Finally, in order to reduce the yield of secondary thermal neutrons, we investigated the inclusion of a thin neutron shield to the MBC setup. A 1.5 cm thick MBC made of polyethylene (G4_POLYETHYLENE) with the same slit dimensions as for the brass MBC was attached to the brass MBC in downstream direction. To analyze the effectiveness of the neutron shield, we added a 'particle filter' to the Geant4 primitive dose scorer, which scored only the dose delivered by neutrons. Apart from the particle filter, the dose scorer properties remained the same as described above.

The primaries were generated in spot-scanning fields of 14 × 14 spots at a lateral distance of 3 mm in both directions (40 × 40 mm$^2$ field size). The pencil beams were modelled as Gaussian in both directions, with standard deviation of 4.7 mm. The beam energy was kept at 180.32 MeV/u as before. The beam was generated directly at the collimator entrance. Per run, $10^7$ primaries were generated.

### 2.3. Particle Identification

The cluster size measured on the CMOS sensor depends on the energy loss of the particle in the sensitive layer of the sensor, enabling to disentangle the primary ion tracks from the ones of the lighter fragments [20]. In a first step, the reference measurement without collimator was analyzed to measure the amount of produced fragments from the material in front of the sensors (beam nozzle and scintillator plastic).

In Figure 3, the distribution of the cluster size of each particle averaged over the first three MIMOSA-28 sensors (front tracker) is shown, where the main peak represents the primary carbon ions. For simulated data, the contribution from secondaries and primaries is also depicted separately for better understanding. The several peaks in the left region are caused by the lighter fragments, and the secondary peak on the right side of the primary peak is due to the pile-up inside the sensors. For analysing the fragments generated inside the MBC, only the primary carbon ions impinging on the front tracker were selected by cutting out the fragments coming from the beam nozzle and the scintillator. The two vertical lines in the plot represent the cut between primaries and secondaries (dashed light green line, cluster size of 29) and that between single and double hits (dashed dark green line, cluster size of 60), respectively. For the back tracker, secondaries generated in the MBC were identified as those tracks for which the mean cluster size was smaller than 33 in simulation, accounting also for the energy loss of the particles in the front tracker. For the experiment, this threshold was varied between 33 and 38, according to the position of the minimum between the distributions for larger and smaller cluster sizes.

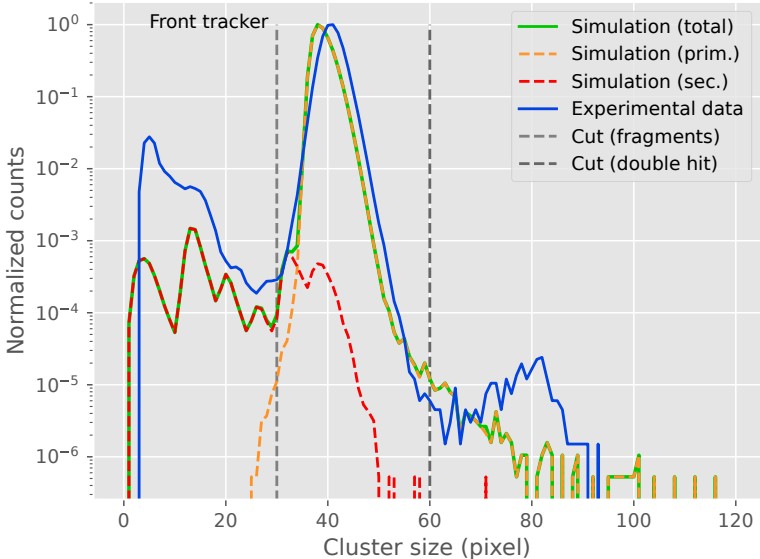

**Figure 3.** Mean cluster size of the front tracker. For the simulated data (green line), the contribution of secondaries (dashed red line) and primaries (dashed orange line) are shown separately. The two vertical dashed lines represent the cut between fragments and primaries (light gray), and between single and double hits (dark gray).

Due to the small sensor size, only 91.3% of the particles reach the back tracker. From a run without MBC, the fragmentation due to the beam nozzle and the scintillator plastic was estimated to be 1.8% of the overall events in experiment. In addition, the fragmentation of the primary carbon ions in the front tracker and the air was found to be 0.6%.

## 3. Results

### 3.1. Measurements

Figure 4a shows the fluence map of the measurements for the best alignment of the collimator. In front of the MBC ($z < 0$), the fluence profile was extrapolated from straight-line fits to the front tracker events; after the MBC, from straight line fits to the rear tracker hits. The broadening of the collimated tracks with distance from the MBC can be clearly observed. In addition, a shoulder in the beam profile develops, starting from ∼20 mm behind the collimator, which dilutes again to a more or less homogeneous underground after ∼70 mm distance to the collimator exit.

Figure 5a shows the broadening of the slit profile in the fluence map as a function of distance to the MBC exit. The broadening is expressed in the standard deviation of a

Gaussian fit to the fluence profile of the central slit in Figure 4b–d. Right at the MBC exit, the fluence profile full width at half maximum was 452 μm, comparable to the slit width. The full width at half maximum became greater than the slit width after a drift of 35 mm. In Figure 5b, the peak-to-valley fluence ratio is shown for the central slit as a function of distance to the MBC exit. The peak-to-valley fluence ratio decreased from ∼90 at the MBC exit to below 40 after a drift of 40 mm. Note, while correlated, the peak-to-valley fluence ratio does not directly translate to PVDR, which was investigated with the extended simulation setup.

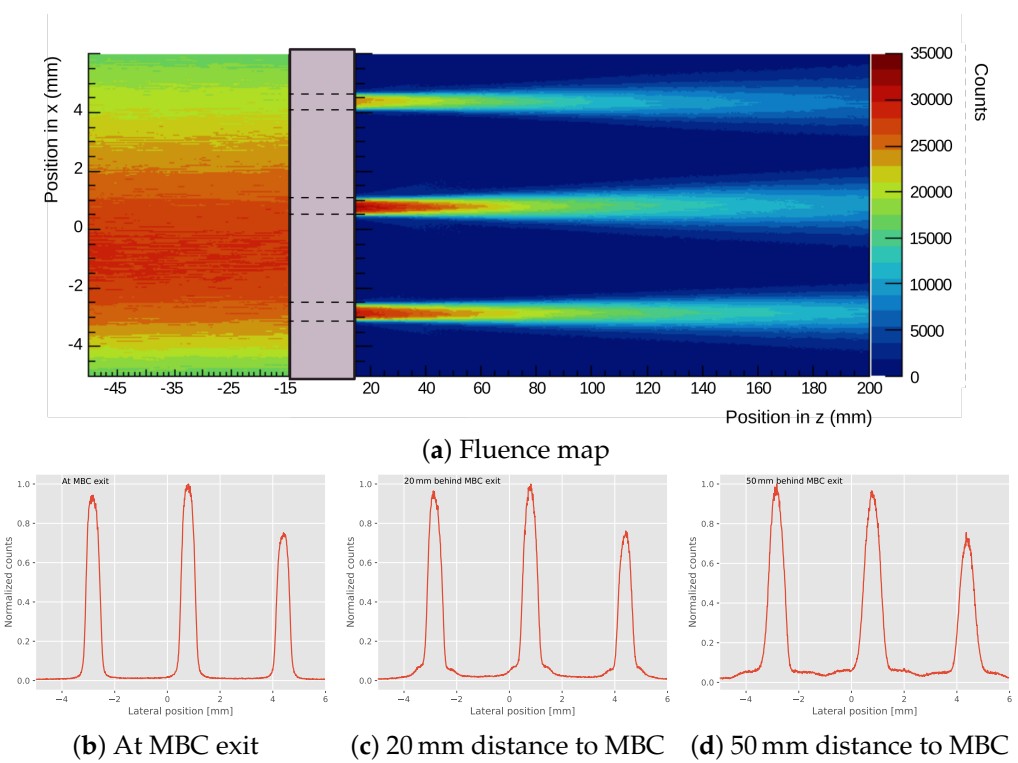

(**a**) Fluence map

(**b**) At MBC exit      (**c**) 20 mm distance to MBC      (**d**) 50 mm distance to MBC

**Figure 4.** (**a**) Fluence map extrapolated as straight line fits from the front ($z < 0$) and from the rear ($z > 0$) tracker setup, respectively, for the experimental data from the measured MIMOSA-28 sensor hits when the MBC was best aligned. (**b**–**d**) Projected lateral profiles of the experimental fluence map at different distances to the MBC exit.

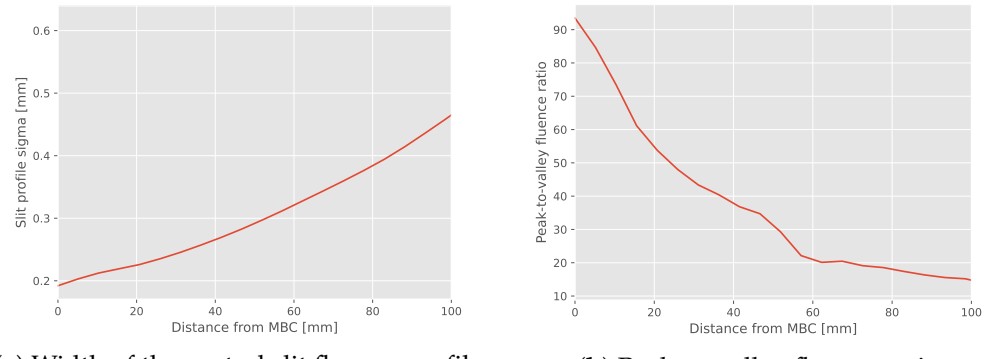

(**a**) Width of the central slit fluence profile      (**b**) Peak-to-valley fluence ratio

**Figure 5.** (**a**) Width of the fluence profile of the central slit, expressed in the standard deviation of a Gaussian fit to the slit fluence profile, as a function of distance from the MBC exit. (**b**) Peak-to-valley fluence ratio for the central slit.

In Figure 6, the 2D distributions of the primary and secondary fluences on the first sensor behind the MBC are shown. The primary fluence follows the slits, but there was

also a small peak in the primary fluence at the valley region. From analyzing the fluence map, this can be attributed to the scattered primaries that originally made up the shoulder in the fluence profiles for closer distances to the MBC exit.

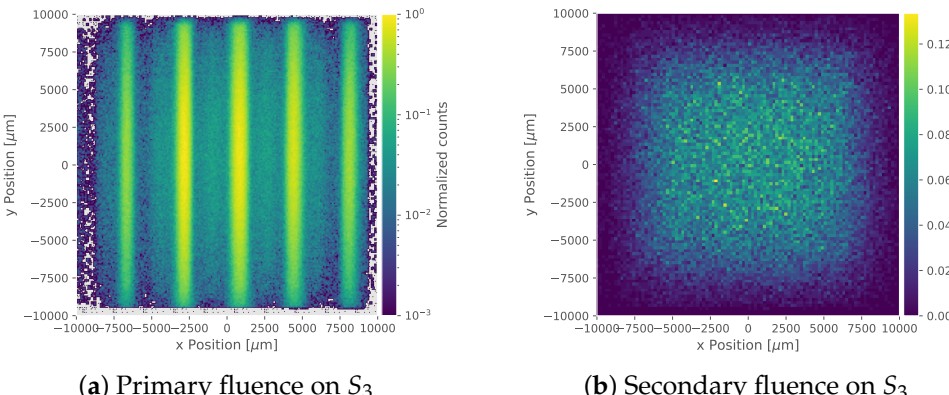

(**a**) Primary fluence on $S_3$                                            (**b**) Secondary fluence on $S_3$

**Figure 6.** Comparison of primary fluence (**a**) and secondary fluence (**b**) on the first sensor behind the MBC in beam direction ($S_3$). The fluence was normalized to the maximum count. For better visibility, the primary fluence (**a**) was plotted with a logarithmic color scale.

In Figure 7, the fluence profiles behind the MBC are shown for the ideal alignment of the MBC. Figure 8 shows the same for a $-0.4°$ rotational misalignment with the beam. With increasing rotational misalignment, the fluence profile deteriorates, presenting an asymmetric shoulder on the left hand side for counter-clock-wise rotations and on the right hand side for clock-wise rotations. At the same time, the fragmentation background also increases, reducing the PVDR. In Figure 9, the ratio of primary and secondary components in the beam is shown, indicating an increase in secondary particle fluence with increasing rotational misalignment of the MBC and a loss of primary fluence, as expected. These measurements were also used to identify the best alignment of the collimator when processing the experimental data, as the position at the primary fluence peak (here defined as $0°$). The figure also enables to estimate the output factor of the MBC at the best alignment, which was 17.8% for the primary fluence in the experiment, and 17.0% in the simulation.

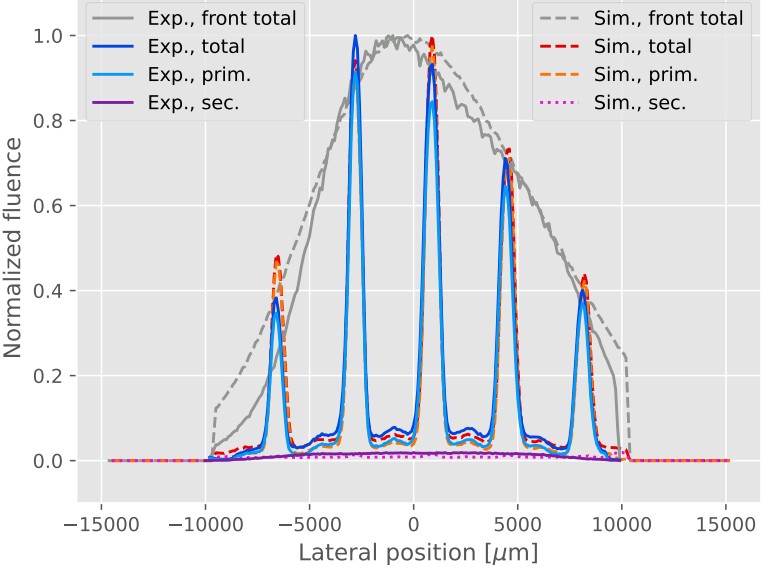

**Figure 7.** Lateral fluence profiles on the first sensor after the MBC for ideal alignment of the MBC. For the experiment, the ideal alignment was determined from the primary/secondary fluence ratio. Simulations are shown as dashed, experiments as solid lines.

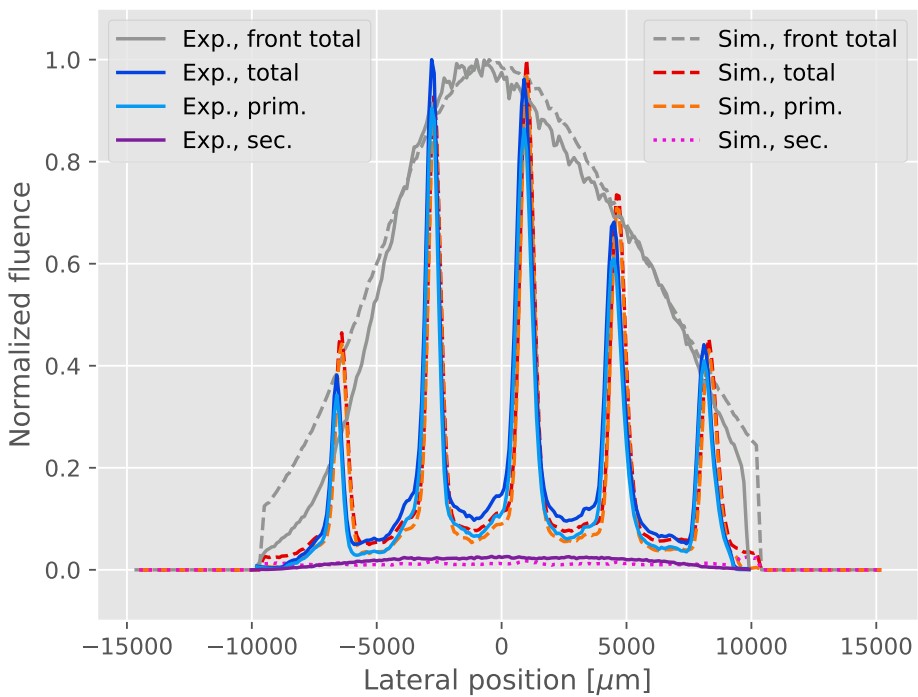

**Figure 8.** Lateral fluence profiles on the first sensor after the MBC for a $-0.4°$ degree rotational misalignment of the MBC. Simulations are shown as dashed, experiments as solid lines.

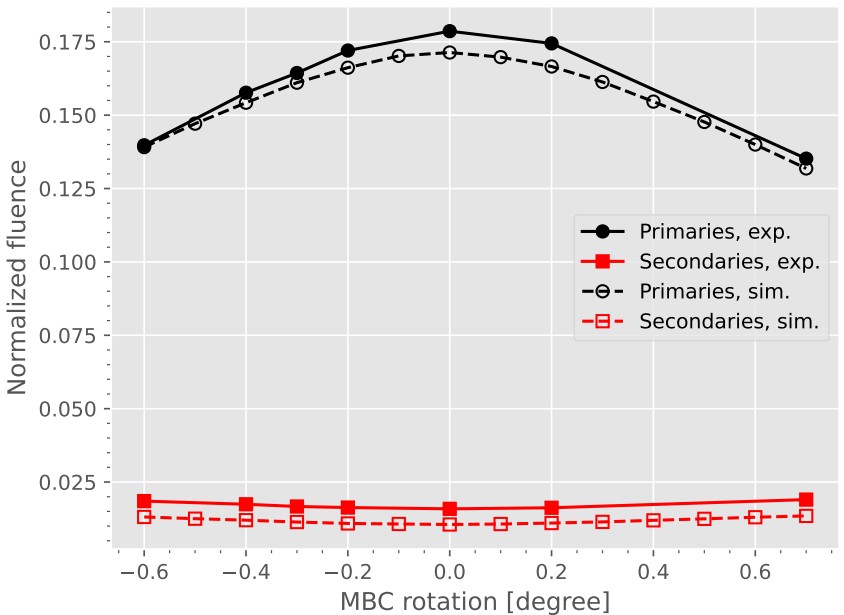

**Figure 9.** Relative primary and secondary particle fluence measured on the back tracker, normalized to fluence of primary ions on the front tracker for different rotational alignments of the MBC. The solid lines show the experimental, the dashed lines the simulated data.

In Figure 10, the distribution of average rear tracker cluster sizes is shown. Apart from larger cluster sizes, in the center of the MBC slits, the average rear tracker cluster sizes also peaked right in the center between MBC slits. This indicates a high LET beam component both in the fluence peaks and valleys. In contrast, the average front tracker cluster size was homogeneous in lateral position on the front tracker. Cluster sizes were in general slightly larger in the experiment, but the general trend was similar compared to the simulation.

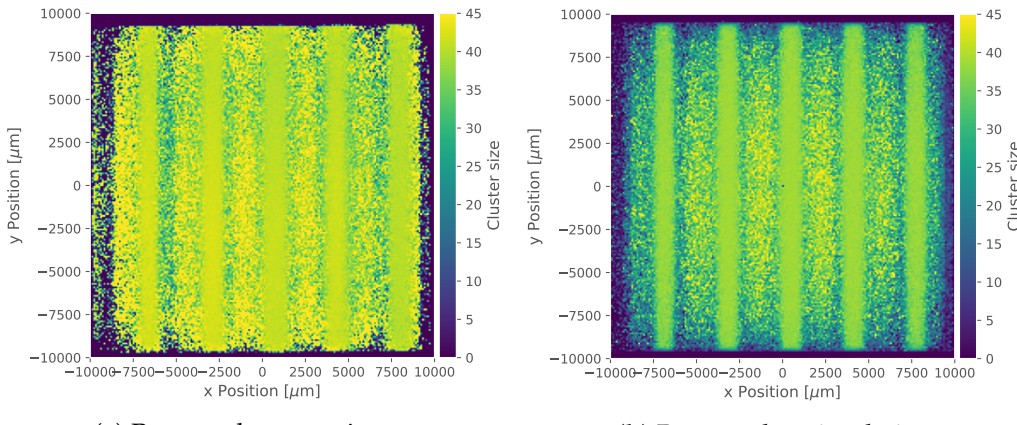

(**a**) Rear tracker experiment        (**b**) Rear tracker simulation

**Figure 10.** Distribution of the mean cluster size measured in the rear tracker based on the particles' position on the first sensor of the rear tracker for the tracked particles. All particles corresponding to the primary ion distribution on the front tracker were considered.

### 3.2. Expanded Parameter Analysis

In this section, the results from the expanded Monte Carlo setup (Section 2.2.2) are presented. Further results are provided in Appendix A.

Dose to Water

Figure 11a shows a 2D view of the simulated dose to water, when the MBC was placed 20 mm in front of the water phantom and was irradiated with a grid of $14 \times 14$ pencil beams at a lateral distance of 3 mm ($40 \times 40$ mm$^2$ field size). The pencil beams were modelled as symmetric Gaussian beams with standard deviation of 4.7 mm. The beam divergence was assumed to be 2 mrad. Figure 11b shows the resulting mean PVDR for the nine central slits and their surrounding valleys. The peak value was calculated as the dose value in the central voxel of each of the slits. The valley dose was evaluated as the value in the dose grid voxel positioned in the middle between two adjacent slits. The PVDR was averaged over 1 mm steps in depth. It can be observed that the PVDR shows a short build up, reaching a PVDR of ~45 between 10 mm and 20 mm depth, and decreases to ~25 shortly before the Bragg peak location. The PVDR shows a strong peak, coinciding with the Bragg peak location. In the bottom row of Figure 11, lateral dose profiles at different depths are depicted, normalized to the maximum dose at that depth. Figure 11c shows the profile at the water phantom entrance, Figure 11d at 35 mm depth, and Figure 11e at 70 mm depth shortly before the Bragg peak. Strong edge scattering effects can be observed right at the water phantom entrance in the form of shoulders in the slit dose profiles. These wash out with increasing depth and only slight shoulders can be observed at a depth of 35 mm, while no visible shoulders are present in the profile at 70 mm depth.

In Figure 12, a comparison of the PVDR between different beam angular divergence and distances between the MBC and the water phantom can be observed. The irradiation parameters, except stated explicitly, were kept the same as in Figure 11. In Figure 12a, the effect of increasing beam angular divergence on the PVDR can be viewed, where larger beam angular divergence reduced the PVDR. In Figure 12b, the effect of increasing distance between the MBC exit and the water tank entrance is depicted for a fixed beam angular divergence of 2 mrad, corresponding to the value at the Marburg Ion-Beam Therapy Center. With increasing distance, the PVDR at the water tank entrance suffered considerably. Especially, the initial maximum in PVDR shifted towards the phantom entrance, until the PVDR monotonously increased from a PVDR of only 10 at the entrance to roughly 20 shortly before the Bragg peak for the 50 mm distance between the MBC exit and the water phantom entrance.

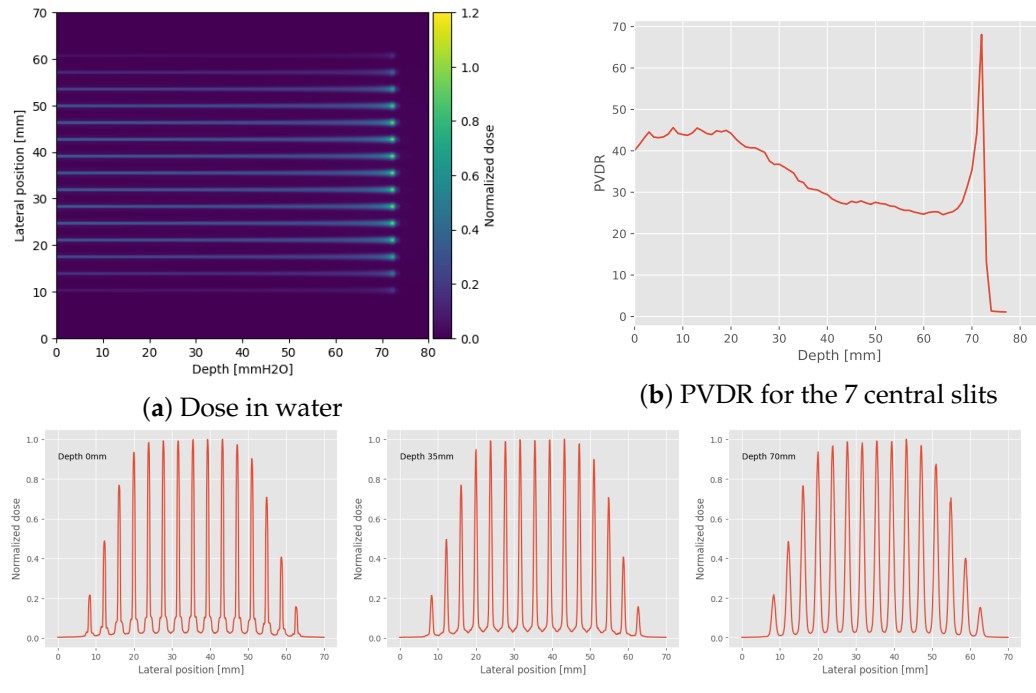

(**a**) Dose in water      (**b**) PVDR for the 7 central slits

(**c**) Profile at 0 mm depth    (**d**) Profile at 35 mm depth    (**e**) Profile at 70 mm depth

**Figure 11.** Top: (**a**) physical dose in water for a $4 \times 4\,cm^2$ rectangular carbon ion field ($14 \times 14$ raster spots) impinging on the brass MBC placed at a distance of 20 mm before the water tank, obtained with the expanded simulation setup. The dose was normalized to the maximum. (**b**) peak-to-valley dose ratio (PVDR) for the 3 central slits and their surrounding valleys as function of depth in the water phantom. Bottom: lateral dose profiles for different depths in the water phantom, normalized to the maximum at that depth. (**c**) At the entrance of the water phantom, (**d**) at 35 mm depth, and (**e**) at 70 mm depth.

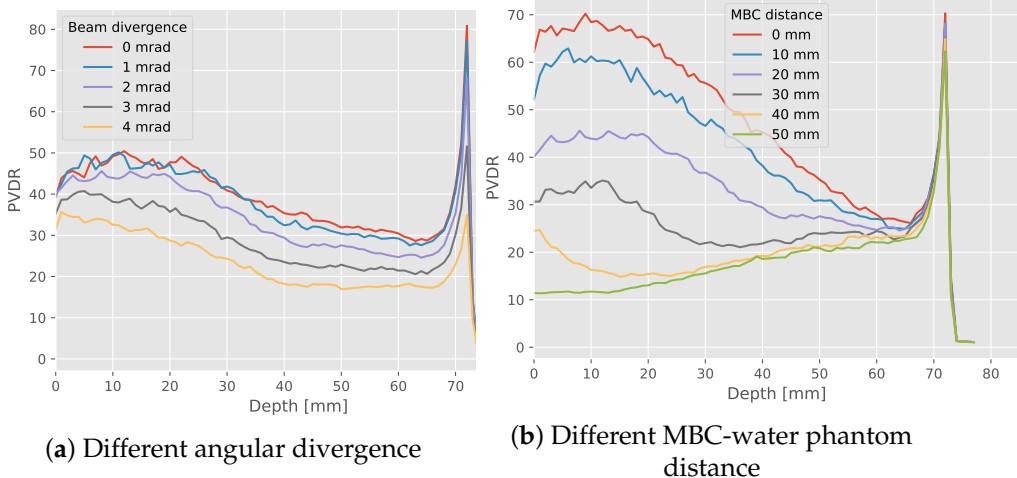

(**a**) Different angular divergence      (**b**) Different MBC-water phantom distance

**Figure 12.** Comparison of the PVDR for different beam and setup parameters from the expanded simulation setup. For (**a**), the distance between MBC and water phantom entrance was kept at 20 mm. For (**b**), the angular divergence was kept at 2 mrad.

### 3.3. MBC Modifications

In Figure 13, the effect of chamfered slit edges on the MBC dose profiles is shown. Figure 13a shows a 2D difference profile to the irradiation with an ideal MBC. The beam angular divergence was kept at 2 mrad, and the distance between the MBC exit and water tank entrance was 20 mm. The chamfered edges result in more pronounced shoulders at

the base of the slit dose profiles (Figure 13c). This also had a negative effect on the PVDR (Figure 13b), which was lower for the chamfered edges compared to the ideal ones.

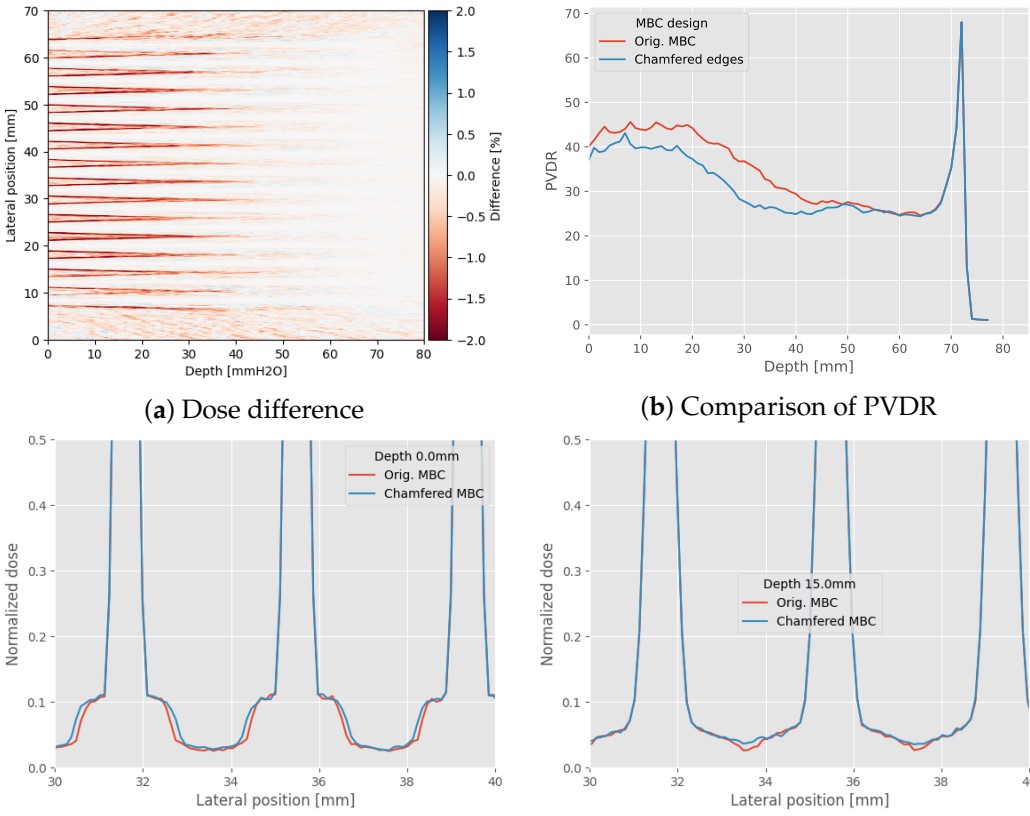

(**a**) Dose difference

(**b**) Comparison of PVDR

(**c**) Profile comparison at 0 mm depth

(**d**) Profile comparison at 30 mm depth

**Figure 13.** Comparison between the ideal MBC with sharp slit edges and a version where the slit edges were chamfered, performed with the expanded simulation setup. Top: (**a**) relative dose difference between the irradiation with the ideal MBC and one with chamfered slit edges based on the expanded Monte Carlo simulation setup; (**b**) comparison of the resulting PVDR for the MBC with and without chamfered edges. Bottom: (**c**) comparison of the dose profile at 0 mm depth in the water tank, zoomed into the base of the slit dose profile; (**d**) the same for a depth of 30 mm.

In Figure 14, the neutron depth dose profile is shown for the original MBC setup, and the setup, including an additional polyethylene neutron shield, was attached to the MBC. The neutron dose was recorded by the same scorer as the total dose but with an added 'particle filter' for neutrons. The dose was then laterally integrated to produce the data in Figure 14a. Again, the beam angular divergence was kept at 2 mrad, and the distance between MBC exit and water tank entrance was 20 mm.

While the neutron dose with the neutron shield exceeded that without neutron shield up to 5 mm depth, the additional neutron shield lowered the neutron dose up to 29% for deeper depths. However, the additional low-density material also increased the washout of the scattered beam components at the MBC edges, as observed in the comparison of the lateral beam profiles. This also affected the PVDR, as observed in Figure 14d. Interestingly, the PVDR was increased for depths between 30 mm and 60 mm in the water tank when compared to the standalone MBC without the neutron shield. In the depth dose profile of the total dose, a secondary peak appeared, which corresponded to primaries that crossed the polyethylene.

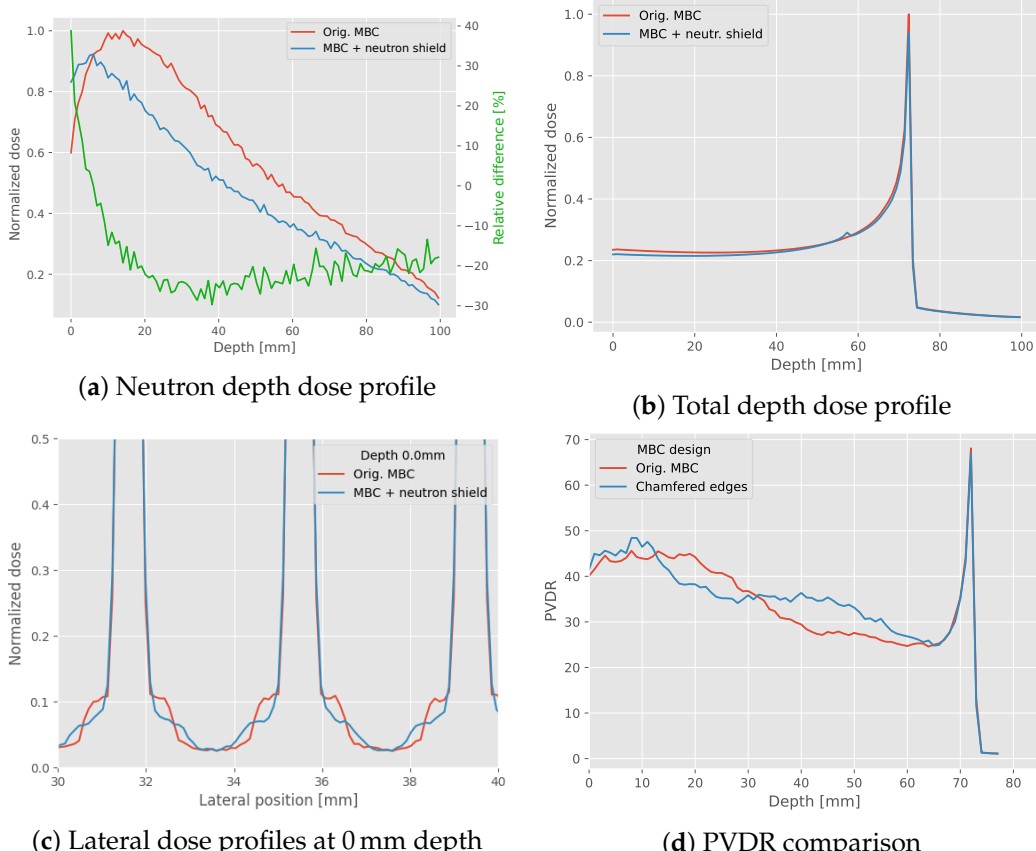

**Figure 14.** (**a**) Comparison of the neutron depth dose profiles between the original MBC setup and one with an additional polyethylene neutron shield placed after the MBC, obtained with the expanded simulation setup. Both dose profiles were normalized to the maximum of the original depth dose profile. (**b**) Comparison of the total depth dose profile with and without neutron shield. (**c**) Comparison of the lateral dose profiles at the entrance of the water tank. (**d**) Comparison of the PVDR.

## 4. Discussion

In this work, we characterized a brass MBC for pre-clinical experiments with carbon ion minibeam therapy. The work comprised two parts: first, we performed in-depth characterization measurements of the MBC at the Marburg Ion-Beam Therapy Center, employing a tracker setup composed of six CMOS monolithic active pixel sensors. In the second part, based on the experience from measurements, we conducted an extended parameter analysis, investigating the delivered dose to water in Monte Carlo simulations.

Due to their high spatial resolution, the MIMOSA-28 sensors are well suited for precision particle fluence measurements [21], similar to those required for characterizing the fluence profile at the MBC slits. This corroborates previous works on X-ray minibeam therapy [18] and pMBRT [19], which already indicated promising potential of CMOS technology for providing high-resolution dosimetry in minibeam experiments. The CMOS sensor high resolution revealed a shoulder at the base of the slit lateral fluence profiles at small distance to the collimator exit (Figure 4d), stemming from edge scattering effects at the corners of the MBC slits. With increasing distance from the MBC, these shoulders were washed out due to the divergence of the collimated beams. Similar shoulders were observed in simulation, both when replicating the measurement setup and in the expanded parameter analysis.

Further, it was shown that applying a 1 mm chamfer to the collimator edges resulted in a broadening of the shoulders at the base of the dose profile and a considerable reduction in PVDR. Whether this also has any significant effect on the results of pre-clinical experiments

remains to be determined, considering the absolute dose difference between the MBC with and without chamfered edges was rather small. Wear and tear of the collimator, which causes a slight chamfer of the collimator edges for the brass MBC, is, therefore, not expected to cause a considerable effect. Still, if the highest manufacturing precision is desired, especially also in terms of each slit's internal alignment, precision manufacturing techniques, such as 3D printing or precision casting, may be applied to manufacture the collimator. The use of 3D steel printing technology would also potentiate more complex MBC designs.

When comparing the cluster size distributions between the simulation and experimental data of the CMOS sensor setup, the peak cluster size was larger by a few pixels in the experiment than in the simulation. This can be attributed to several contributing factors: firstly, the beam energy distribution before the nozzle beam monitors was not known and rather estimated to be Gaussian with a standard deviation of 1‰ of the energy. Further, the beam nozzle monitors were modelled as water slab targets of low density, which modelled the overall energy loss and scattering of the beam nozzle. Rather than choosing a slab of water with thickness corresponding to the water thickness of the beam nozzle, the enlarged low-density water slab target also takes into account the correct length for beam drift inside the nozzle. However, this neglects the effect of high-density materials, such as the tungsten wires of the multi-wire proportional chamber position detectors. These high-Z-target materials introduce slight differences in the energy loss and fragmentation inside the beam nozzle compared to simulating the nozzle as a water box. The contribution of fragmentation inside the nozzle can be viewed as negligible in the context of this work. A further factor affecting the cluster size comparison was that in the simulation the energy loss in the MIMOSA-28 sensor's sensitive layer was scored directly and then converted to cluster size using the analytical formula by Reidel et al. [20]. This neglects any possible uncertainties introduced by noise or the cluster extraction software. Most importantly, however, in the simulation, all secondary particles were recorded individually, and their energy loss was not added to that of the primary ions. As such, any possible larger cluster sizes due to close hits were treated as separate clusters. Nonetheless, the overall trend of the cluster size distribution observed in Figure 10 was very comparable in simulation and experiment.

All of the above contributes also to the observed differences between primary and secondary fluence, which were obtained by applying similar cuts in experiment and in simulation. Nevertheless, considering the uncertain modelling of nuclear fragmentation in Monte Carlo for carbon ions [30,31], the agreement between simulation and experiment was deemed satisfactory for the presented analysis.

Due to the thin material budget of the MIMOSA-28 sensors' sensitive layer, the recorded cluster size is approximately proportional to the LET of the incoming particles. We investigated the 2D distribution of the average cluster sizes for the MBC-collimated carbon ion beams, which is proportional to the track-averaged LET. Here, it was observed that a high LET beam component is present not only in the peaks but also in the valley centers, which was similar for both simulation and experiment. A detailed simulation-based analysis of secondary fragment contribuition for oxygen and carbon ion minibeam therapy in the literature [15] investigated spread-out Bragg peak carbon ion minibeam therapy with a setup similar to the one used in this work. One of their key findings was that, for a ctc of 3500 μm, the valley dose was dominated by nuclear fragments, while the on average shorter-range $\delta$-electrons did not contribute considerably. This is in agreement with the higher track-averaged LET in the valley center indicated by the cluster size distribution in our work. This high LET component may also be relevant in the context of pre-clinical dosimetry with detectors that are sensitive to LET quenching. Radiosensitive films, for example, are known to increasingly underestimate the dose with increasing LET [32]. The high LET component both in the peak and valley, therefore, means film measurements of both regions are affected by quenching. While previous work has demonstrated good

agreement between film and and micro-diamond dosimetry [8], care should be taken when evaluating iMBRT with radiosensitive films.

As part of the experiments, we performed a robustness analysis for the rotational alignment of the MBC. Similar to what has already been observed in the work by Ortiz, De Marzi, and Prezado [14] for protons, increasing rotational misalignment introduced a shift in the peak position of the slit dose, as well as an asymmetric slit profile with a shoulder on one side. This resulted in a decrease in the PVDR and also the depth dose profile (see Appendix A). We also analysed the fluence of primary and secondary particles on the rear tracker, as determined from the measured mean cluster size, to determine the best alignment of the collimator. The ratio between primary and secondary particles presented a clear maximum at the best rotational alignment of the MBC. Since this observation is expected to be independent of the MBC general shape, in future experiments, the analysis of primary and secondary fluence could be used for precise collimator positioning, possibly even for more unconventional MBC designs, as made possible through 3D printing technology.

In order to further study the effect of different beam and setup parameters on the carbon ion minibeams generated with the MBC, the experimentally validated simulation framework was expanded to enable assessing the dose to water for pencil beam scanning fields. This simulation framework was kept as user friendly as possible, allowing several interesting parameters to be set directly from the command line without further changes needed to the code. Especially, the code supports simple exchange of MBC geometry by utilizing the stl format for CAD drawings.

Using the extended parameter simulations, the PVDR as function of depth in a water tank was assessed. Here, PVDR values as high as ~45 were reached for shallow depths for the parameter values close to those at the Marburg Ion-Beam Therapy Center with realistic distance between MBC and water tank. The PVDR reached even ~70 if no distance between the MBC and the water tank was assumed. The PVDR of 45 is, however, smaller than the PVDR reported for previous carbon ion minibeam experiments by Martinez-Rovira et al. [8], possibly related to the thicker (7 cm thick) tungsten MBC employed in their work. The study by Martinez-Rovira et al., as well as a simulation study by Gonzalez, Peucelle, and Prezado [15], also reported a general decrease in the PVDR with increasing depth. Here, the PVDR reached its minimum of ~25 (for the realistic setup in Figure 11) at a depth of 65 mm and then increased again at the Bragg peak region, where the peak dose increase outweighed the valley dose increase from the broadening of the minibeams. This may be because the previous works studied spread-out Bragg peaks composed of several energy layers, whereas, here, only one energy layer was investigated. The observed PVDR increase at the Bragg peak is expected to wash out for spread-out Bragg peaks or when a ripple filter is used in the setup. Further differences compared to Gonzalez's work may reside in differences in the Monte Carlo simulation settings. Nonetheless, the PVDR values observed in this work were at a comparable level in this and previous works.

With the extended simulations, the negative effect of increasing beam angular divergence and distance between the MBC exit and water tank entrance on the PVDR was shown. While increased blurring of the slit dose profile with increasing angular divergence is expected, Figure 12a highlights the sensitivity of the setup to this parameter. Of course, extending the MBC thickness would reduce the impact of larger angular divergences by favoring parallel carbon ion tracks even more. However, it would reduce the output factor of the MBC, requiring a larger initial dose to achieve the same peak dose. In general, the slit width and ctc did not result in a homogenization of the dose at the peak depth for the chosen energy and available beam angular divergence with the Marburg Ion-Beam Therapy Center carbon ion beam. The intended use case with this collimator was shoot-through iMBRT experiments with small animals, conducted at the GSI medical experiment cave. Smaller ctc would be needed if a homogeneous dose at the peak was desired with just one treatment direction. For example, Gonzalez, Peucelle, and Prezado [15] showed that a ctc of 0.98 mm leads to a homogeneous dose in the center of the investigated spread-out Bragg peak (PVDR $\leq$ 1.15) but also led to a reduction in the PVDR in the entrance region by a

factor of 100. Smaller ctc also increases the likelihood of carbon ions crossing through the slit walls and exiting through a neighbouring slit. An alternative to a reduced ctc would be interlaced minibeams [33,34], where multiple minibeam fields would be used to achieve a homogeneous tumor dose while preserving the low valley doses.

The MBC should be positioned as close as possible to the irradiated object, since the PVDR is highly sensitive to increasing distance as seen in Figure 12b. The distance not only affects the increase of the width of the slit dose profile due to the drift in air, but also affects the shoulder in the fluence profile resulting from the edge scattering (see Figure 4c,d). During experiments, the positioning of the MBC needs to be kept consistent to avoid related uncertainties. Small MBC to target distances may therefore be challenging, especially for small animal experiments, due to the necessity of regularly changing part of the setup. Experiment designs should consider secure positioning of the MBC. Due to the considerable difference observed in the PVDR, re-evaluation of the MBC position may be advisory for long experiments with many setup changes.

In order to reduce secondary neutron dose, the extended simulation setup was used to assess a novel neutron shielding concept for the MBC. Due to the high Z of the collimator material, production of secondary neutrons inside the MBC is unavoidable. This would particularly also be the case for even heavier ions than carbon, which have recently gained renewed interest in iMBRT [12]. We demonstrated how a thin polyethylene neutron shield attached to the collimator in downstream direction may be useful to reduce the dose from secondary neutrons. The neutron shield thereby had the same slit design as the primary MBC and can be 3D-printed for highest precision. Already, the 1.5 cm additional polyethylene neutron shield added in this work achieved an up to 29% reduction in the neutron dose (compare Figure 14a). However, additional optimization of the neutron shield slit design is needed to avoid the dose disturbance from primary carbon ions that traversed the polyethylene, which introduces increased scattering and range pull-back. For example, the neutron shield slits could be designed in a delta shape, taking the angular distribution of the primary carbon ions at the MBC exit into account. There is also a trade-off between neutron shield thickness and MBC patient distance that needs to be considered.

In general, the sensitivity of the MBC to setup uncertainties renders MBCs an inferior choice for iMBRT generation when compared to beam focusing methods [13,35]. However, since no contemporary clinical carbon ion therapy facility has access to the required beam optics to achieve the desired minibeam widths, pre-clinical experiments need to rely on MBCs as a more practical option. The insights on MBC-generated carbon ion minibeams provided by this work will help to improve future experiments and lay the basis for advanced MBC designs tailored to the desired experimental outcome. The fully characterized MBC of this work has already been used in pre-clinical minibeam experiments at our institution.

Finally, due to the variable RBE of carbon ions, the inclusion of biological modelling for characterizing the minibeams in terms of their biological effect would be desired. This may change the PVDR considerably. While we observed a high LET component in the dose valleys in this work (see Figure 10), the corresponding particle fluence and doses are rather small, such that the effect on the RBE is not clear. Due to the large variations between RBE models [36,37], an in silico study of the biological effectiveness of carbon ion minibeams should ideally employ multiple RBE models. Ultimately, the RBE of carbon ion minibeam therapy needs to be confirmed in pre-clinical experiments, such as those currently underway.

## 5. Conclusions

This work presented an in-depth characterization of an MBC currently in use for carbon ion minibeam therapy pre-clinical experiments. For the first time, the usefulness of CMOS pixel sensors for MBC characterization in terms of not only lateral particle fluence but also LET and secondary particle identification was demonstrated. Similar to what has been demonstrated in the literature for protons, correct positioning of the MBC is crucial to ensure

accurate and reliable experimental outcomes. A simulation framework was developed for studying the parameter space of not only carbon but, in general, iMBRT, and was utilized to demonstrate the effect of different beam and setup parameter choices. In the future, this framework will aid in improving experimental designs for MBC-generated iMBRT. As such, a novel MBC setup featuring a neutron shield manufactured from polyethylene was proposed and demonstrated to reduce the dose delivered by secondary neutrons.

**Author Contributions:** Conceptualization, L.V., C.-A.R., Y.P. and C.G.; methodology, L.V., C.-A.R., C.S. and C.G.; software, L.V., C.-A.R. and C.S.; validation, L.V., C.-A.R. and C.S.; formal analysis, L.V. and C.-A.R.; investigation, L.V. and C.-A.R.; resources, U.W., C.S., C.G. and M.D.; data curation, L.V. and C.-A.R.; writing—original draft preparation, L.V.; writing—review and editing, L.V., C.-A.R., C.S., Y.P., M.D., U.W. and C.G.; visualization, L.V. and C.-A.R.; supervision, U.W., M.D. and C.G.; project administration, M.D. and C.G.; funding acquisition, Y.P., M.D. and C.G. All authors have read and agreed to the published version of the manuscript.

**Funding:** This project has received funding from the European Union's Horizon 2020 research and innovation programme under grant agreement No 101008548 (HITRIplus). Y.P. received funding from the European Research Council (ERC) under the European Union's Horizon 2020 research and innovation program (grant agreement no. 817908).

**Data Availability Statement:** All data will be made available upon reasonable request to the corresponding author. The simulation code is freely available at git.gsi.de/lennart.volz/minibeams.

**Acknowledgments:** We want to acknowledge the Marburg Ion-Beam Therapy Center for providing beam time and technical support for the presented experiments.

**Conflicts of Interest:** The authors declare no conflict of interest.

## Appendix A

### Appendix A.1. Single Slit Dose Profile Broadening

Figure A1 shows the broadening of the full width half maximum of the dose profile of the central slit as obtained for the setup shown in Figure 11 in the main text. The full width half maximum was computed by interpolating the slit dose profile. Starting at a full width half maximum of ∼0.6 mm at the entrance of the water tank, the slit dose profile broadens to ∼1.1 mm shortly before the Bragg peak.

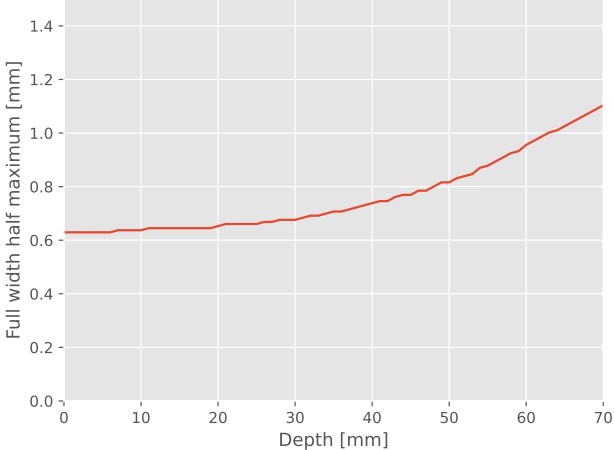

**Figure A1.** Full width half maximum as function of depth for the dose profile of the central slit from the setup shown in Figure 11 in the main article.

### Appendix A.2. Dose Difference between Aligned and Rotated MBC

Figure A2 shows a comparison between the dose to water when the MBC was perfectly aligned with the beam and when it was rotated by $0.4°$. The rotation not only affects the PVDR negatively but also the depth dose profile. Due to the lateral shift in peak and valley

positions, the PVDR was calculated by finding the maxima and minima at each depth instead of using the slit center position and center between slits as done for the main results.

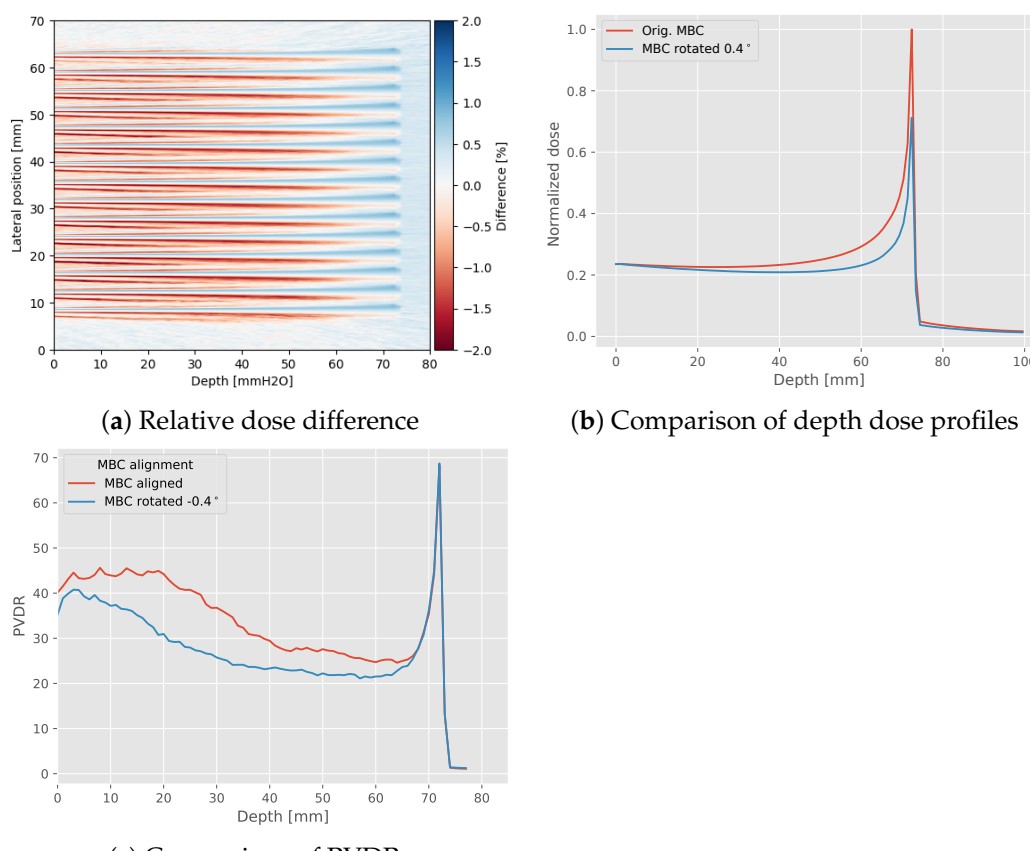

(**a**) Relative dose difference

(**b**) Comparison of depth dose profiles

(**c**) Comparison of PVDRs

**Figure A2.** Comparison of an irradiation with the MBC perfectly aligned with the beam and one where it was rotated by 0.4° from the expanded simulation setup. (**a**) Relative dose difference; (**b**) comparison of depth dose profiles integrated over both lateral directions; (**c**) comparison of PVDRs.

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
