# Peer review of "Investigating Slit-Collimator-Produced Carbon Ion Minibeams with High-Resolution CMOS Sensors"

_instruments, doi:10.3390/instruments7020018_

Round 1

Reviewer 1 Report

Well elaborated study. The only point the authors should discuss in more detail how the lateral dose profiles may alter when reducing the ctc in order to come to more practical applications of this study.

More comments are directly made in the attached text

see attached file

Author Response

We thank the reviewer for their assessment of our manuscript. Please find attached a point-by-point reply with the associated changes made to the manuscript.

Reviewer 2 Report

This is an interesting, well written paper that was a pleasure to read. I believe the following small revisions will improve the paper.

Sec 2.1.1 - Please describe the minimum detectable energy threshold for the MIMOSA sensor and (importantly) what this corresponds to in terms of the LET and energy deposition of secondary particles.

Sec 2.2.2 - Please describe the physics cut used in the Geant4 simulation and the reason for this choice.

Figure 2 - The choice of color in this figure made (my printed copy at least) rather hard to read. The authors should review the use of colors in this figure, especially the bright yellow used for simulation (sec).

Figure 4 - It was very hard for me to interpret the differences between these figures, and the authors might consider including a single large version of this figure instead.

Line 337 - 'Nevertheless, considering the anyways' - use of anyways not correct English

Author Response

We thank the reviewer for their assessment of our manuscript. Please see the attached point-by-point reply with the changes made to the manuscript. 
